# Diabetes-Induced Cardiac Autonomic Neuropathy: Impact on Heart Function and Prognosis

**DOI:** 10.3390/biomedicines10123258

**Published:** 2022-12-15

**Authors:** Susumu Z. Sudo, Tadeu L. Montagnoli, Bruna de S. Rocha, Aimeé D. Santos, Mauro P. L. de Sá, Gisele Zapata-Sudo

**Affiliations:** 1Programa de Pós-Graduação em Medicina (Cirurgia Geral), Faculdade de Medicina, Universidade Federal do Rio de Janeiro, Rio de Janeiro 21941-902, Brazil; 2Programa de Pesquisa em Desenvolvimento de Fármacos, Instituto de Ciências Biomédicas, Universidade Federal do Rio de Janeiro, Rio de Janeiro 21941-902, Brazil; 3Programa de Pós-Graduação em Cardiologia, Instituto do Coração Edson Saad, Universidade Federal do Rio de Janeiro, Rio de Janeiro 21941-913, Brazil; 4Faculdade de Enfermagem, Universidade Federal do Estado do Rio de Janeiro, Rio de Janeiro 22290-180, Brazil; 5Instituto do Coração Edson Saad, Faculdade de Medicina, Universidade Federal do Rio de Janeiro, Rio de Janeiro 21941-913, Brazil

**Keywords:** cardiac autonomic neuropathy, diabetes, oxidative stress, inflammation, endothelium dysfunction

## Abstract

Cardiovascular autonomic neuropathy (CAN) is a severe complication of the advance stage of diabetes. More than 50% of diabetic patients diagnosed with peripheral neuropathy will have CAN, with clinical manifestations including tachycardia, severe orthostatic hypotension, syncope, and physical exercise intolerance. Since the prevalence of diabetes is increasing, a concomitant increase in CAN is expected and will reduce quality of life and increase mortality. Autonomic dysfunction is associated with reduced baroreflex sensitivity and impairment of sympathetic and parasympathetic modulation. Various autonomic function tests are used to diagnose CAN, a condition without adequate treatment. It is important to consider the control of glucose level and blood pressure as key factors for preventing CAN progression. However, altered biomarkers of inflammatory and endothelial function, increased purinergic receptor expression, and exacerbated oxidative stress lead to possible targets for the treatment of CAN. The present review describes the molecular alterations seen in CAN, diagnosis, and possible alternative treatments.

## 1. Introduction

Diabetes mellitus (DM) has become one of the leading causes of premature death in most countries, mainly through increased risk of cardiovascular diseases, which are responsible for 50 to 80% of deaths in diabetic patients [1]. The number of diabetic patients is increasing due to population growth and aging, the progressive prevalence of obesity, and sedentary lifestyle. In the period 1980–2014, there was an increase from 108 to 422 million DM diagnoses, and the prevalence has increased more rapidly in low- and middle-income countries [2]. In 2017, DM overall incidence, prevalence, disability-adjusted death, and years of quality life were 22.9; 476.0; 1.37, and 67.9 million, respectively. Additionally, it is estimated that in 2025, DM will affect 26.6; 570.9; 1.59, and 79.3 million, respectively [3]. Progression of DM is associated with chronic complications including cardiomyopathy, retinopathy, nephropathy, and neuropathy. Diabetic neuropathy provokes an autonomic nervous system dysfunction with nervous fibers damaged, which causes cardiovascular autonomic neuropathy (CAN). This review includes an overview of the DM-induced CAN regarding its definition, epidemiology, molecular mechanisms, and therapeutics.

## 2. Pathogenesis

Changes in insulin concentration and activity observed in DM promote abnormalities in lipid metabolism, favoring microvascular and macrovascular complications [4]. Lipid accumulation, oxidative stress, and inflammation cause activation of stress-sensitive kinases with consequent inhibition of insulin signaling [5]. These changes in organs sensitive to the action of insulin develop renal, neuronal, and cardiac comorbidities. DM cause mitochondrial and endoplasmic reticulum dysfunctions in cardiomyocytes, which lead respectively to oxidative stress and abnormal Ca^2+^ handling, which are responsible for death and rigidity of myocytes and inducing diabetic cardiomyopathy, respectively [6]. Impairment of cytosolic Ca^2+^ flux in DM cardiomyocytes can decrease diastolic and systolic cardiac function, promoting ventricular dysfunction in the absence of coronary artery disease or hypertension. High levels of blood glucose cause vascular and neural damage because glucose in the cytosol is directed into the polyol pathway, in which it is converted to sorbitol by the enzyme aldose reductase, using the nitric acid adenine dinucleotide phosphate (NADPH) as a cofactor. The excessive consumption of NADPH culminates in its depletion, resulting in the generation of reactive oxygen species (ROS) [7]. ROS stimulates the oxidation of low-density lipoprotein (LDL) and ox-LDL, which are not recognized by the LDL receptor and are captured by macrophages, causing the formation of foam cells and atherosclerotic plaques, which leads to vascular disorders associated with DM [8]. Excessive ROS production results in increased oxidative stress and provokes hyperglycemia-induced damage through an increase in advanced glycation end product (AGE) formation, which results from the glycosylation of lipids, lipoproteins, and amino acids, promoting their deposition in the sub-endothelial layer and inducing endothelial dysfunction [9]. In fact, AGEs can directly inactivate the endothelium nitric oxide (eNO), which determines the lack of endothelium-dependent vasodilation [10]. The activation of the gene-binding receptor (RAGE) induces the production of ROS, which activates NF-kB, causing multiple pathological changes in gene expression [11]. Several factors could explain the pathogenesis of diabetic cardiomyopathy, including cardiac metabolic disorders, subcellular signaling abnormalities, autonomic dysfunction, activated renin-angiotensin-aldosterone system, inflammation, oxidative stress, and maladaptive immune response [7]. The inflammatory response involves a complex cascade of events involving many types of cells. Inflammatory signaling in cardiomyocytes usually occurs in an early phase of response to myocardial injury and implies overproduction of mitochondrial ROS [12]. The activation of several signaling pathways, such as NF-κB, c-jun NH_2_-terminal kinase, or p38-MAPK, can mediate the state of inflammation, which is linked to insulin resistance, playing an important role in diabetic complications. NF-kB represents one of the most important mediators of the inflammatory process because its activation is associated with increased release of cytokines, such as tumor necrosis factor alpha (TNF-α)—which is often involved in cardiac damage contributing to cardiac hypertrophy and fibrosis—as well as ventricular dysfunction in the diabetic heart [13]. 

CAN characterized by an imbalance between sympathetic and parasympathetic activity is a major complication of DM [14]. The predominance of sympathetic activity can exacerbate cardiomyopathy in diabetic patients because of the increase of release of myocardial catecholamines, activation of adrenergic receptors, and activation of the renin–angiotensin–aldosterone system (RAAS) [15] (Figure 1). 

CAN is classified according to its progress: 1. subclinical stage—characterized by lack of symptoms but with abnormal simpatovagal balance, decreased baroreflex sensitivity; [16,17] 2. early stage—tachycardia at rest; 3. advanced stage—exercise intolerance, cardiomyopathy with left ventricular dysfunction, orthostatic hypotension, and silent myocardial ischemia [18,19].

## 3. Diagnosis of DM-Induced Cardiac Autonomic Neuropathy

As the prevalence of DM increases, a concomitant increase in CAN occurs, which cannot be diagnosed in the early stage, also occurs [18,20]. DM can induce neuron injury to the autonomic nervous system, which causes autonomic dysfunction involving initially the parasympathetic and later the sympathetic system. Although the mechanism of neuronal dysfunction is not completely clear, hyperglycemia, which is responsible for the increase of ROS and AGE, leads to the damage of peripheral nerves affecting initially the vagus nerve [21]. Long-term progression of DM is associated with lack of blood pressure response to sleep or exercise, indicating damage to the sympathetic system, which is correlated with high mortality. The severity of CAN depends on glucose level, time of DM, blood pressure, and aging-induced neuronal death [22]. Initially, altered HRV occurs, and it is followed by resting tachycardia. The next step of the sympathetic predominance consists of decrease in coronary flow and consequent cardiac dysfunction. The progression of CAN reveals a denervation of sympathetic vasomotor producing an inadequate vascular and HR response combined with orthostatic hypotension. 

Diagnosis involves the use of cardiac autonomic reflex tests (CARTs), including the evaluation of blood pressure and heart rate variability (HRV) during different maneuvers. HRV consists of spontaneous and induced fluctuations in heart rate (HR—very low, low, and high frequency) [23]. The spectral analysis of the balance between sympathetic and vagal activities is most sensitive in early stages of CAN. It is essential to perform HRV and the gold standard tests (deep breathing, Valsalva, and orthostatic) to achieve early diagnosis of this complication of DM [24]. 

### 3.1. Diagnostic Tests for CAN

#### 3.1.1. Autonomic Testing 

The gold standard tests of CARTs involve the measurement of changes in HR and BP to physiological maneuvers [25]. The standard CARTs recommended for diagnosis of CAN include: the deep breathing (DB) test, the Valsalva maneuver (VM) test, the lying-to-standing (LS) test at 30:15, and the BP response to orthostasis [25,26]. 

-Deep breathing (DB) test

The DB test is performed by measuring beat-to-beat HRV in a supine position. Patient respiration is paced at six breaths per minute by a metronome or similar device during electrocardiogram (EKG) recording. The expiration-to-inspiration (E/I) index is calculated as the ratio between the three maximum and the three minimum RR intervals in a respiration cycle [25]. Minimal threshold for normal E/I ratios depend on patient age and decline from 1.17 at 20–24 years to 1.02 at 70–75 years [26]. Differences in HR under 10 bpm and E/I ratios lower than the threshold are considered abnormal [26].

-HR response to Valsalva maneuver

The patient is asked to perform forced exhalation (to 40 mmHg in a manometer) for 15 s with simultaneous EKG recording. Healthy subjects develop tachycardia during exhalation and bradycardia after release. The ratio of the longest RR interval after exhalation and the shortest RR interval during exhalation is calculated, and normal values should be 1.2 or higher [25,26].

-Lying-to-standing (LS) test

During continuous EKG monitoring, the patient changes from a supine position to standing up. The longest RR interval between the 25th and 35th beat and the shortest RR interval between the 10th and 20th beat after change of posture are measured [25]. Values under 1.03 and ratios of 30:15 are considered abnormal [26].

-Blood pressure (BP) response to orthostasis

BP is measured initially in the supine position and 2 min after standing up. A reduction in BP of between 10–29 mmHg or over 30 mmHg is indicative of borderline or abnormal BP response [25,26]. Moreover, a decrease of over 20 mmHg in systolic or over 10 mmHg in diastolic BP is also used as being indicative of orthostatic incompetence [25].

#### 3.1.2. HR Variability (HRV) Measures Used for Diagnosis

HRV is determined as the variations in consecutive RR intervals and may be analyzed using long (usually 24 h) or short (<10 min) EKG measurements, although the latter may be preferred for investigation purposes. Direct analysis provides HRV parameters in the time domain, mainly standard deviations (SDNN) or root mean squared deviations (RMSSD) of successive RR intervals [25]. Frequency–domain analysis is performed after data transform (e.g., fast Fourier transform) to obtain HRV spectra. These histograms are divided in low (LF) and high frequency (HF) regions, which provide information on the influences of both autonomic branches on HR. The ratio LF/HF is mainly used as an index of autonomic balance, generally referred as the sympathetic-to-parasympathetic ratio. Although a reduction in the HF band may reflect a vagal dysfunction in early CAN, it may be accompanied by reduction in LF, indicating concomitant dysregulation of sympathetic tone [25,26].

A confirmed diagnosis of CAN consists in the presence of three altered tests and the presence of associated orthostatic hypotension reflects greater severity and poor prognosis. DM causes autonomic neuropathy, which is the disability of the compensatory autonomic mechanism to regulate blood pressure. The increase in vascular resistance in response to the activation of baroreceptors induced by hypotension does not occur because of failure of the sympathetic reflex in diabetic patients. For that reason, patients have orthostatic hypotension as a result of damage to the efferent sympathetic vasomotor fibers. More than 50% of diabetic patients diagnosed with peripheral neuropathy will have CAN. When in an advanced condition, ischemia and asymptomatic myocardial infarction can occur, as can left ventricular dysfunction, which is linked to increased risk of renal failure and sudden death [27]. Sudden cardiac death can also be a consequence of the arrhythmogenic effect, which promotes atrial and ventricular arrhythmias [14]. Therefore, arrhythmias can manifest in early stages of CAN during the decreased parasympathetic activity phase. Among diabetic patients, 38–44% may suffer from cardiac dysautonomia with prognostic implications and higher cardiovascular mortality. There are slight differences in relation to risk factors between DM type-1- or type-2-induced CAN, leading to a multifactorial approach in its prevention [28]. Thus, an intensive treatment of hyperglycemia added to a multifactorial approach is beneficial in patients with DM and can result in the delay and prevention of CAN.

## 4. Molecular Mechanisms of DM-Induced Cardiac Autonomic Neuropathy

DM-induced macrovascular changes can lead to coronary artery disease, atherosclerosis, and heart failure, but microvascular changes cause peripheral vascular dysfunction, stroke, nephropathy, polyperipheral neuropathy, and retinopathy [19]. Inflammation plays an important role in the development of those complications resulting from DM. The pro-inflammatory environment resulting from hyperglycemia and/or insulin resistance is the result of the higher expression of inflammatory mediators, such as IL-1β, IL-6, TNF-α, TGF-β, IKKβ, and JNK, that may have local or systemic action. Inflammatory mediators act on the endothelium so that these cells alter their transcription profile, promoting a greater expression of cell recognition markers (ICAM-1 and VCAM-1) and facilitating the diapedesis of hematopoietic cells (lymphocytes and monocytes). Released cytokines and chemokines can activate other cells in addition to macrophages, antigen presenting cells (APC), effector cells (Th1, Th2), or memory cells (lymphocyte B). This can occur in the heart, promoting an immune response that perpetuates inflammation, leading to an oxidative and hypoxic environment. These events promote autophagy, apoptosis, necrosis, non-effective neovascularization, hypertrophy, collagen deposition, and fibrosis, which—in the long term—cause cardiac dysfunction and failure. Briefly, the mitochondrial overproduction of ROS, which leads to usually irreversible tissue damage, such as apoptosis, hyperplasia, hypertrophy, remodeling, or fibrosis, alters the glycolytic metabolism, increasing the use of glucose by routes other than ATP production [29].

Additionally, diabetic peripheral neuropathy is caused by the effects of chronic exposure to hyperglycemia, which promote glycotoxic stress, inducing increased oxidative stress with increased ROS; dysfunction of the antioxidant system; activation of the JNK, AP-1, PKC, NFκB, and MAPK; the additional formation of polyols and glycosamines; AGE favoring the transcription of genes encoding TGF-β, IL-1, IL-6, and TNF-α; and induced nitric oxide (iNOS). ROS induces depression of autonomic ganglion synaptic transmission, contributing to increased risk of fatal cardiac arrhythmias as well as to sudden death after myocardial infarction [29]. Mitochondrial dysfunction is associated to all these processes that lead to apoptosis and autophagy, bioenergetic dysregulation, neurovascular dysfunction, and inflammation and demyelination of nerves [30].

CAN is considered a diffuse neuropathy characterized as chronic and progressive with altered signaling pathways (polyol, poly (ADP ribose) polymerase (PARP), PKC, AGE) in neuronal cells (Figure 2). 

It also has increased concentration of pro-inflammatory cytokines (IL-1β, IL-6, TNF-α), and exacerbation of oxidative stress. High glucose levels cause the following changes: 1. conversion into sorbitol, whose intracellular increase promotes osmotic stress and greater electrolyte output from the cell, which causes impairment of Schwann cells from peripheral neurons; 2. positive regulation of the hexosamine pathway, resulting in increased N-acetyl glucosamine (GlcNAc) and, consequently, the induction of oxidative stress; 3. exacerbation of oxidative stress due to lipid peroxidation and reduction of glutathione (GSH) levels and enzymes involved in antioxidant defense, such as catalase and superoxide dismutase (SOD); 4. increase inTNF-α levels, which provides, indirectly, the occurrence of neuronal apoptosis; 5. increase in AGE receptors (RAGE) in endothelial and Schwann cells of peripheral nerves; 6. activation of PKC in nerve tissue. Considering the oxidative stress in the vagus nerve (first nerve affected), an increase in malondialdehyde (MDA) levels and reduced levels of GSH, SOD are detected in diabetic animals [31]. 

DM induces autonomic dysfunction since baroreflex sensitivity is decreased and sympathetic and parasympathetic modulation is impaired. Reduced HRV and vagal modulation index are probably consequent to high glycemic levels [15]. Parasympathetic denervation and sympathetic predominance reduce blood pressure and HR, causing exercise intolerance. Increased sympathetic activity is also attributed to upregulated chemoreflex associated with sleep apnea syndrome (intermittent hypoxia) in diabetic patients [32]. 

A correlation between cardiac autonomic control parameters and biomarkers of inflammatory and endothelial function is detected in patients with DM type 2 [33]. Thus, the levels of subclinical inflammation biomarkers, such as IL-6, IL-1beta, and high-sensitivity C-reactive protein, were inversely associated with cardiac vagal control parameters. In contrast, biomarkers of endothelial function, nitric oxide, and nitric oxide endothelial synthase were positively related to cardiac vagal control parameters, suggesting that there is a relationship between inflammation, endothelial dysfunction, and cardiac autonomic dysfunction in DM type 2 [33]. Early investigation of these reported biomarkers is important in interfering with the prognosis of CAN. 

The profiles of non-coding long RNAs (lncRNAs) related to inflammation were demonstrated to be co-expressed with messenger RNA (mRNA) in cervical ganglia after lesion due to DM [34]. The functional target genes of lncRNAs were identified, revealing that the genes interacted with lncRNAs and participate in various functions related to cervical sympathetic ganglia, such as immune response, cell migration, and chemotaxis, indicating that lncRNAs could be potential target for treatment of diabetic patients with CAN or for diagnostic molecular markers. 

Increased P2Y14 purinergic receptor expression is observed in DM, with consequent activation of satellite glial cells [35]. Treatment with short hairpin RNA (shRNA) P2Y14 or naringin reduced the overexpression, indicating that naringin may negatively regulate the expression of the P2Y14 receptor of glial cells in the upper cervical ganglia and can attenuate CAN. In addition, naringin shRNA also decreased changes in HR and blood pressure induced by DM and partially restored the low frequency/high frequency ratio that was increased. Partial recovery of the abnormal sympathetic nerve discharge was promoted via naringin shRNA treatment, which also normalized the levels of ROS and antioxidant factors in DM. In addition to this, the enzyme glucokinase (GCK) is overexpressed in ganglia and may be involved in diabetic cardiac sympathetic neuropathy mediated by the activation of P2X3 receptor. This role was confirmed through electrophysiological study in which the currents activated by the P2X3 agonist increased in ganglia neurons isolated and transfected with GCK plasmid [36]. To investigate the role of the P2X3 receptor in sympathetic nerve activity in diabetic cardiac autonomic neuropathy, shRNA was used, which led to attenuation of increased blood pressure and heart rate as well as normalized heart rate variability [37]. Furthermore, the administration of shRNA P2X3 reduced IL-1β and TNF-α levels in the ganglia and serum level of epinephrine and decreased the phosphorylation level of extracellular regulated proteins 1/2 (ERK1/2), suggesting that the P2X3 receptor could be a therapeutic target of diabetic cardiac autonomic neuropathy.

The increase in angiotensin II (Ang II) levels due to the high genic expression of the angiotensin-converting enzyme (ACE) associated with DM impairs autonomic function, intensifying sympathetic modulation of the heart and reducing its vagal modulation. DM induced in rats could promote significant reduction in HRV, sympathetic dominance, and increased HR and BP with increased cardiac hydroxylase tyrosine activity [38]. Apolipoprotein B, oxidative stress, and inflammatory markers were significantly reduced after treatment with vitamin C. In addition, administration of vitamin C in diabetic rats can restore HRV, suggesting that vitamin C can have effects on the improvement of diabetic CAN [38].

Lipid metabolites of the phosphatidylcholine and sphingomyelin classes were found at elevated plasma levels and linked to cardiac autonomic dysfunction in patients with recently diagnosed type 2 DM but not type 1 DM. This fact indicates that alteration of lipid metabolism occurs in the initial phase of diabetic CAN [39].

## 5. Alternative Treatment of DM-Cardiac Autonomic Neuropathy

Prevention of CAN includes the maintenance of controlled plasmatic glucose levels, treatment of risk factors regulating dyslipidemia and hypertension, and lifestyle intervention. The treatment of CAN relies on the relief of symptoms, but the high mortality can be explained by the imbalance in sympatovagal activity, damage of cardiac sympathetic innervation, impaired vasodilation of coronary arteries mediated by reduced sympathetic activity, deregulation of calcium mobilization, and inflammation [20] (Figure 3). Cardiovascular autonomic reflex tests, including the evaluation of autonomic response using alterations of HR and blood pressure, determine the progression of CAN [20]. 

### 5.1. Symptomatic Treatment

The main non-pharmacological approach to treating DM consists of aerobic exercise and reductions in fat and carbohydrate intake. The Diabetes Prevention Program (DPP) study demonstrated beneficial effects of lifestyle intervention because of reductions in HR, HRV, and QT interval in diabetic patients [40], which could delay the appearance of cardiac complications. However, a study involving 25 non-diabetic patients with metabolic syndrome undergoing a 24-week lifestyle intervention showed a reduction in oxidative stress markers but no alteration in CAN indices [28]. Lifestyle change and exercise are generally indicated and contribute to the improvement of orthostatic hypotension associated with autonomic dysfunction [19,28]. Avoid use of tricyclic antidepressants, diuretics, and α-adrenoreceptor antagonists to reduce possible worsening of orthostatic hypotension. It is important to increase water consumption and the number of small meals and avoid abrupt changes in body posture. The resting tachycardia characteristic of the worsened stage of vagal dysfunction in CAN may be treated with cardioselective (i.e., beta-1) beta-adrenergic antagonists to improve vagal function [19,29,30,41]. As CAN progresses, reduced sympathetic response occurs and provokes orthostatic hypotension [20], which is treated with approved drugs targeted at primary or neurogenic orthostatic hypotension, such as droxidopa, midodrine, clonidine, pyridostigmine, and fludrocortisone. Midrodine, an alpha-1 adrenergic agonist is a vasopressor and fludrocortisone is a mineralocorticoid promoting increase of plasma volume, and both increase blood pressure in non-responsive patients to other therapy [15,30,31]. In advanced CAN, there is a denervated heart because of the abolished parasympathetic and sympathetic activities, with lack of HR change in response to exercise, stress, or sleep [31,42].

### 5.2. Disease Modifying Therapies

Moderate weight reduction associated with caloric restriction could improve autonomic cardiovascular modulation, sympathetic–vagal balance, and reduction of sympathetic tone in patients with metabolic syndrome or obesity [32,43]. The effects of 10% reduction in body mass in overweight or obese patients [33,34,35,36,37] improved autonomic status, HRV, and sympathetic–vagal balance. 

Glycemia and blood pressure control are key factors in preventing CAN progression and should be addressed as main outcomes for therapy [15,31,38]. However, long-term therapy with metformin, an insulin sensitizer, was recently associated with CAN aggravation due to its interference in vitamin B12 absorption in diabetic patients. Vitamin supplementation is recommended for diabetic patients in biguanide therapy for over 5 years or with additional risk factors (advanced age, bariatric surgery, or anti-ulcer therapy) and may be combined with calcium supplementation [41].

Since DM promotes chronic inflammation with high levels of interleukin-6, which has been associated with reduced HRV; treatment with agonist of glucagon-like peptide-1 (GLP-1) has shown anti-inflammatory and neuroprotective effects and could result in the prevention of CAN. Treatment with metformin reduces vascular and systemic inflammation detected by the reduction of TNF-alpha level. Metformin not only reduces adipose tissue inflammation but also oxidative stress in myocardium through the activation of adenosine monophosphate-activated protein kinase. In addition to this, metformin reduces sympathetic predominance of CAN early stage [44,45]. 

GLP-1 receptor agonists cause weight-lowering and glycemic control effects, leading to a reduction in cardiovascular mortality [46]. However, despite preclinical evidence and reduced cardiovascular risk, short-term (3–18 months) therapy with GLP-1 agonists (exenatide 0.15–2.0 mg weekly or liraglutide 1.2–1.8 mg daily) does not improve cardiac autonomic function in diabetic patients but significantly increases HR [25,47]. 

Both sodium–glucose transporter-2 (SGLT-2) and angiotensin-converting enzyme (ACE) inhibitors promote preventive and therapeutic effects in attenuating diabetic cardiac dysautonomia [25,46,47,48,49,50,51]. Despite the lack of SGLT-2 in the heart, studies with gliflozins demonstrated a significant reduction in cardiovascular risk and deaths consequent of DM, which could result in part from the regulation of the autonomic nervous system through the inhibition of renal reabsorption of glucose [47,50,52]. Recently, the EMBODY trial indicated that medium-term (6 months) use of empagliflozin (10 mg p.o. daily) improved most heart rate variability indexes in type 2 diabetic patients taking beta blockers when compared to baseline measurements [47]. Although there is some remaining controversy, the effect of SGLT-2 inhibitors on cardiac autonomic tone is still the object of current research. Another SGLT-2 inhibitor, dapagliflozin, interferes with the cardiac autonomic function measurements (improved HRV) in patients with type 2 DM and cardiac autonomic neuropathy [53]. 

ACE inhibitor controls the systemic inflammation and oxidative stress because its effect of normalizing sympathetic activity. Carvedilol, a non-selective beta antagonist and alpha antagonist, promotes antioxidant activity and when in combination with ACE inhibitor produces vascular relaxation [19].

In a series of small trials in diabetic patients with CAN free of other cardiovascular disease, chronic oral treatment (3–12 months) with quinapril (20 mg daily) increased parasympathetic tone as measured via CARTs, spectral analysis of HR, and LF/HF ratio. Combination with losartan (100 mg daily) showed little further improvement, which suggested that an additional beta blockade by quinapril not seen in other ACE inhibitors could underlie the effect seen in autonomic control of HR [50]. Beta blockers also show some beneficial effects on cardiac autonomic function in diabetic patients with high cardiovascular risk, but the lack of solid evidence restricts their widespread use [47,48,49]. 

Antioxidants, which are scavengers of free radicals, can reduce the formation of ROS and consequently decrease orthostatic hypotension. Antioxidant therapy (alpha-lipoic acid, vitamin E) or C peptide is responsible for better prognosis for CAN. Alpha-lipoic acid inhibits hexosamine and AGE pathways, while vitamins are immune stimulators and inhibitors of ROS-metabolite-induced DNA damage [19,47,48]. Attenuation of CAN progression through vitamin D supplementation shows a U-shaped dose–response relationship, and studies suggest that clinical benefit is achieved only for diabetic patients with insufficient serum levels (<50 nmol/L) [54]. Administration of minocycline—an inhibitor of microglial activation—was also investigated, and its use for 6 weeks demonstrated improvement of CAN in diabetic patients with good tolerability [48]. 

A systematic review [55] evidenced the beneficial effects of aldose reductase inhibitors in initial stages of CAN because they reduce the glucose used in polyol pathway and prevent the deleterious effects of reduced accumulated sorbitol and fructose in neuronal tissue. Oral treatment with epalrestat (50 mg tid) or tolrestat (200 mg qd) in diabetic patients with clinical CAN improved the HRV and postural BP and was devoid of interferences in glycemic control or adverse effects.

Administration of catechin, a phytonutrient of the polyphenol family, controlled body weight loss and reduced plasma glucose levels and cardiac hypertrophy in diabetic animals [31]. In addition, catechin reversed the increased level of MDA and the reduced levels of GSH, catalase, and SOD in the vagus nerve, indicating the contribution of this compound to the reduction of oxidative stress in the DM model. The reduction in Schwann cell proliferation, lymphocytic infiltration, axonal edema, and demyelization of the vagus nerve after catechin administration suggests that this polyphenol attenuated neural damage. Since catechin also improved hemodynamic parameters, it is a promising candidate in the treatment of cardiac diabetic autonomic neuropathy with focus on neural damage reduction.

It is crucial to diagnose and treat CAN early to reduce the morbidity and mortality associated with this chronic disease. For that reason, interventions to reduce adipose tissue inflammation are essential in retarding the progress of CAN.

## 6. Conclusions

CAN etiology is multifactorial and, despite being a common complication of DM, is still under-diagnosed, having a great impact on morbidity and prognosis. CAN is a microvascular complication, the development of which is associated with obesity, smoking, hypertension, and dyslipidemia. Chronic high glucose levels promote oxidative stress and inflammation, which are responsible for the vascular endothelial, cardiac, and neuronal damage. It is essential the prevention of CAN or early and adequate treatment because its progression is associated with the appearance of diabetic nephropathy. In addition to glucose and hyperlipidemia control combined with lifestyle modification, it is essential to interfere with vascular endothelial dysfunction and thrombosis and use specific treatment of orthostatic hypotension induced by oxidative and nitrosative stress. The complexity of CAN’s pathophysiological mechanism contributes to the need to identify new treatment strategies. 

## Figures and Tables

**Figure 1 biomedicines-10-03258-f001:**
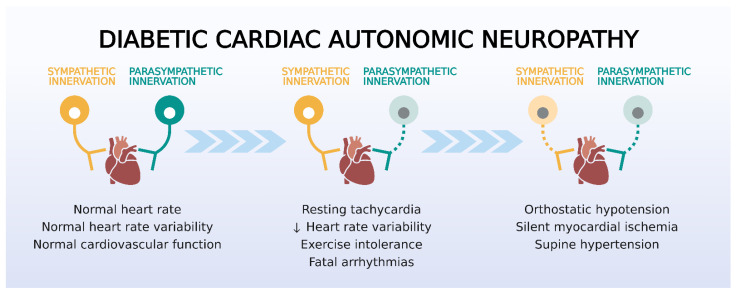
Evolution of symptoms of cardiac autonomic neuropathy (CAN) in diabetes. Patients display reduced variability in heart rate and resting tachycardia in the early phase of parasympathetic dysfunction, which underlie the reduced exercise tolerance and the increased arrhythmogenesis. Later stages of CAN involve sympathetic denervation and cause hemodynamic dysregulation and silent myocardial ischemia. Source: Own authorship.

**Figure 2 biomedicines-10-03258-f002:**
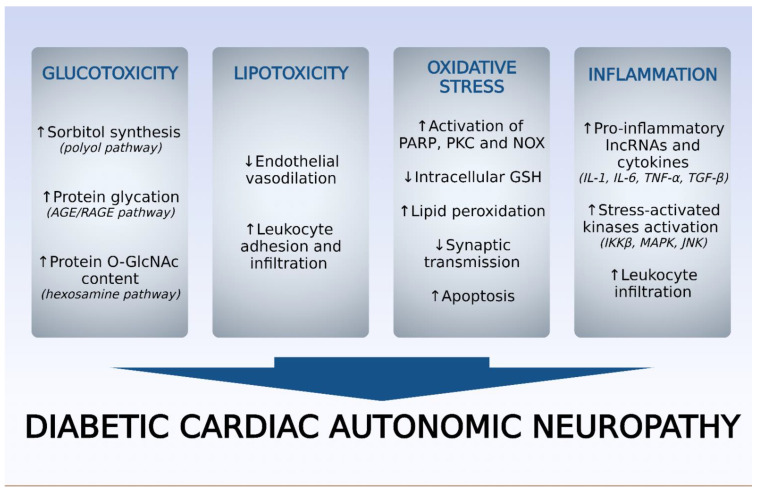
Pathophysiological factors underlying the development and aggravation of diabetic cardiac autonomic neuropathy (DCAN). AGE, advanced glycation end-products; RAGE, receptor for advanced glycation end-products; GlcNAc, *N*-acetyl glucosamine; PARP, poly(ADP-ribose) polymerase; PKC, protein kinase C; NOX, reduced nicotinamide adenine dinucleotide phosphate oxidase; GSH, reduced glutathione; IL, interleukin; TNF-α, tumor necrosis factor-α; TGF-β, transforming growth factor-β; IKKβ, inhibitor of nuclear factor kappa B kinase; MAPK, mitogen-activated protein kinase; JNK, c-Jun N-terminal kinase. Source: Own authorship.

**Figure 3 biomedicines-10-03258-f003:**
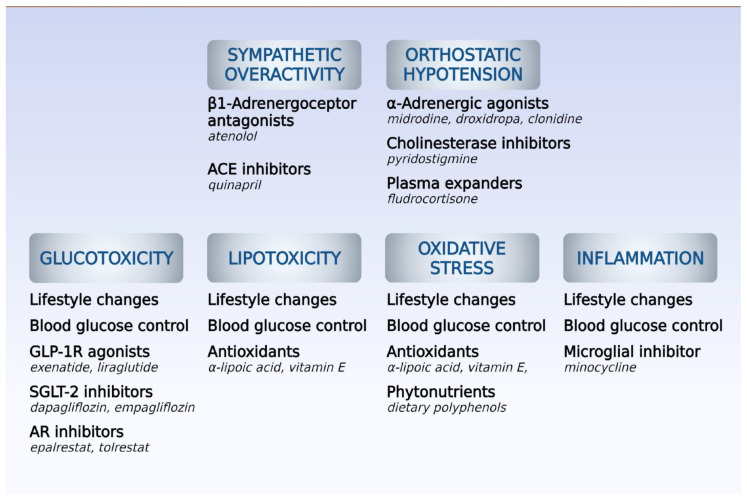
Pharmacological strategies for management of main causes and symptoms of diabetic cardiac autonomic neuropathy (DCAN). ACE, angiotensin-converting enzyme; GLP-1R, receptor for glucagon-like peptide-1; SGLT-2, sodium/glucose cotransporter 2; AR, aldose reductase. Source: Own authorship.

## Data Availability

Not applicable.

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
