# Peer review of "Diabetes-Induced Cardiac Autonomic Neuropathy: Impact on Heart Function and Prognosis"

_biomedicines, 2022, doi:10.3390/biomedicines10123258_

Round 1
Reviewer 1 Report
1) Rewrite and extensive grammar check required through the manuscript.
2) In introduction section, provide recent statistics on increasing trend of diabetes.
3) Reduce introduction to provide an overview of the manuscript and prepare a separate section to provide a detailed pathophysiology of CAN.
4) Provide the classification of CAN.
5) In section 2 ‘Epidemiology and Diagnosis’, no epidemiology is provided, and section 2 reads as very similar to introduction.
6) Diagnosis section doesn’t include ‘blood pressure responds to sustained handgrip’. Is it not first line of diagnosis test?
7) In section 3, lines 138 -178 are majorly based on review articles. Are these studies specific to CAN? If no, then clarify how are these statements linked to CAN.
8) Diabetes is often associated with abnormal (decreased) blood flow. Are any such observations known to be linked with CAN?
9) Figure 2 is missing non-coding long RNAs.
Author Response
Responses to Reviewer 1:
1) Rewrite and extensive grammar check required through the manuscript.
A native speaker colleague edited the manuscript.
2) In introduction section, provide recent statistics on increasing trend of diabetes.
As suggested by the reviewer, introduction has been altered as follows:
Page 2:
Diabetes mellitus (DM) has become one of the leading causes of premature death in most countries, mainly through increased risk of cardiovascular diseases, which are responsible for 50 to 80% of deaths in diabetic patients [1]. The number of diabetic patients is increasing due to population growth and aging, the progressive prevalence of obesity and sedentary life style. In the period of 1980-2014, there was an increase from 108 to 422 million DM diagnosis, and the increase of prevalence is more rapidly in low- and middle-income countries [2}. In 2017, DM overall incidence, prevalence, disability-adjusted death and years of quality life (DALYs) were 22,9; 476,0; 1,37 and 67,9 million, respectively. And, it is estimated that in 2025, those parameters will alter to 26,6; 570,9; 1,59 and 79,3 million [3]. Progression of DM is associated with chronic complications including cardiomyopathy, retinopathy, nephropathy and neuropathy. Diabetic neuropathy provokes an autonomic nervous system dysfunction with nervous fibers damaged, which causes the cardiovascular autonomic neuropathy (CAN). This review includes an overview of the DM-induced CAN regarding its definition, epidemiology, molecular mechanisms and therapeutics.
3) Reduce introduction to provide an overview of the manuscript and prepare a separate section to provide a detailed pathophysiology of CAN.
Authors reduced the introduction and it was added a section containing pathogenesis.
4) Provide the classification of CAN.
The following paragraph was added to the manuscript:
Page 3:
CAN is classified according to its progress: 1. subclinical stage - characterized by lack of symptoms but with abnormal simpatovagal balance and decreased baroreflex sensitivity; [16,17] 2. early clinical stage - resting tachycardia because of sympathetic tone increase; 3. Advanced clinical stage - exercise intolerance, cardiomyopathy with left ventricular dysfunction, orthostatic hypotension, and silent myocardial ischemia [18,19].
5) In section 2 ‘Epidemiology and Diagnosis’, no epidemiology is provided, and section 2 reads as very similar to introduction.
Subtitle was altered to Diagnosis…..and text related to epidemiology was placed in the Introduction. Authors removed the phrases containing similar remarks.
Page 4:
Since the prevalence of DM is increasing, a concomitant increase in CAN occurs, which diagnosis is not done in early stage [20, 21]. DM can induce neuron injury to the autonomic nervous system, which causes autonomic dysfunction involving initially the parasympathetic and later the sympathetic system. Although the mechanism of neuronal dysfunction is not completely clear, hyperglycemia, which is responsible for the increase of ROS and AGE, leads to the damage of peripheral nerves affecting initially the vagus nerve [22]. Long-term progression of DM is associated with lack of blood pressure response to sleep or exercise indicating damage of sympathetic system, correlated to high mortality. The severity of CAN depends on the glucose level, time of DM, blood pressure and aging-induced neuronal death [23]. Initially, occurs altered HRV and followed by resting tachycardia. The next step of the sympathetic predominance consists of decrease of coronary flow and consequent cardiac dysfunction. The progression of CAN reveals a denervation of sympathetic vasomotor producing an inadequate vascular and HR response combined with orthostatic hypotension.
6) Diagnosis section doesn’t include ‘blood pressure responds to sustained handgrip’. Is it not first line of diagnosis test?
The gold standard tests used to assess CAN are the cardiovascular autonomic reflex tests (CARTs) which did not include the blood pressure response to sustained handgrip in recent guidelines. This parameter is not effective factor for the diagnosis of CAN, but could be a predictor of risk to develop type 2 DM and cardiovascular events. Consequently, the results of this test direct to introduce prevention strategies. (Setor K. Kunutsor, Ari Voutilainen, Jari A. Laukkanen. Handgrip strength improves prediction of type 2 diabetes: A prospective cohort study. Annals of Medicine, 2020; DOI: 10.1080/07853890.2020.1815078).
The sensitivity and specificity of sustained handgrip test, in the CAN diagnosis, is 24 and 79%, respectively and does not show any association with the others tests included in CARTs (Körei et al. Why Not to Use the Handgrip Test in the Assessment of Cardiovascular Autonomic Neuropathy among Patients with Diabetes Mellitus? Curr Vasc Pharmacol. 2017;15(1):66-73. doi: 10.2174/1570161114666160822154351).
On the other hand, the blood pressure response to handgrip is a prognostic factor because it is useful in the management of diabetic patients with high risk of cardiovascular outcomes (Morikawa et al. Handgrip Strength Is an Independent Predictor of Cardiovascular Outcomes in Diabetes Mellitus. Int Heart J 2021;62(1):50-56. doi: 10.1536/ihj.20-677).
7) In section 3, lines 138 -178 are majorly based on review articles. Are these studies specific to CAN? If no, then clarify how are these statements linked to CAN.
This section describes initially the molecular mechanisms involved in the macrovascular complications induced by DM. In the following paragraph, authors continue to report the factors involved in DM-induced neuropathy. Finally, in the next paragraph, it was described the pathways which play important role in the development of CAN. Authors considered that this sequence of information regarding the DM-induced CAN could ensure better understanding by readers.
8) Diabetes is often associated with abnormal (decreased) blood flow. Are any such observations known to be linked with CAN?
Oxidative stress and metabolic disorder can promote vascular damage resulting in dysfunction of nerves inducing neuropathy. The hyperglycemia-induced vascular dysfunction and ischemia are important factors to the development of diabetic neuropathy (Nukada, H. Ischemia and diabetic neuropathy. Handb Clin Neurol 2014;126:469-87. doi: 10.1016/B978-0-444-53480-4.00023-0). In addition to this, persistent hyperglycemia is associated with increased formation of AGEs, which can result in atherosclerosis, an additional factor to endothelial dysfunction. Glycocalyx disruption consequent to hyperglycemia and oxidative stress causes capillary dysfunction, which provokes reduction of nerve conduction velocity because of nerve damage (Ostergaard et al. The effects of capillary dysfunction on oxygen and glucose extraction in diabetic neuropathy. Diabetologia volume 58, pages666–677, 2015). Altered endothelial nitric oxide synthesis is a central factor to neuronal abnormalities during metabolic syndrome and is associated with changes in vascular dynamic performance parameters such as blood pressure, blood flow velocity, blood flow. CAN is significantly associated with the presence of retinopathy, which suggests that an impairment of autonomic peripheral blood flow control might be a contributing factor in the formation of microvascular lesions (Valensi et al. Factors involved in cardiac autonomic neuropathy in diabetic patients. J Diabetes Complications 1997 May-Jun;11(3):180-7).
9) Figure 2 is missing non-coding long RNAs.
It was altered Figure 2 to include the missing factor.
Reviewer 2 Report
The authors reviewed DM-induced cardiac autonomic neuropathy. However, I have some comments.
1) Please describe the diagnostic criteria of HRV and the gold standard tests (deep breathing, Valsalva and orthostatic) for cardiac autonomic neuropathy in details.
2) Are there any differences in the prevalence and characteristics of cardiac autonomic neuropathy between IDDM and NIDDM?
3) A thorough review by a native English speaker would be helpful.
Author Response
Responses to Reviewer 2:
1) Please describe the diagnostic criteria of HRV and the gold standard tests (deep breathing, Valsalva and orthostatic) for cardiac autonomic neuropathy in details.
As suggested by the reviewer, authors described the diagnostic tests in the following paragraphs:
- Autonomic testing for diagnostic of CAN
The gold standard tests of CARTs involve measurement of changes in HR and BP to physiological maneuvers [25]. The standard CARTs recommended for diagnosis of CAN include: the deep breathing (DB) test, the Valsalva maneuver (VM) test, the lying-to-standing (LS) test at 30:15, and the BP response to orthostasis [25,26].
1.1. Deep breathing (DB) test
The DB test is performed by measuring beat-to-beat HRV in supine position. Patient respiration is paced at six breaths per minute by a metronome or similar device during electrocardiogram (EKG) recording. The expiration-to-inspiration (E/I) index is calculated as the ratio between the 3 maximum and the 3 minimum RR intervals in a respiration cycle [1]. Minimal threshold for normal E/I ratios depend on patient age and decline from 1.17 at 20-24 years to 1.02 at 70-75 years [26]. Differences in HR under 10 bpm and E/I ratios lower than threshold are considered abnormal [26].
1.2. HR response to Valsalva maneuver
Patient is asked to perform a forced exhalation (to 40 mmHg in a manometer) for 15 seconds with simultaneous EKG recording. Healthy subjects develop tachycardia during exhalation and bradycardia after release. The ratio of the longest RR interval after exhalation and the shortest RR interval during exhalation is calculated and normal values should be 1.2 or higher [25,26].
1.3. Lying-to-standing (LS) test
During continuous EKG monitoring, patient changes from supine position to standing up. The longest RR interval between the 25th and 35th beat and the shortest RR interval between the 10th and 20th beat after change of posture are measured [1]. Values of 30:15 ratios under 1.03 are considered abnormal [26].
1.4 Blood pressure (BP) response to orthostasis
BP is measured initially in the supine position and 2 minutes after standing up. A reduction in BP between 10-29 mmHg or over 30 mmHg are indicative of borderline or abnormal BP response [25,26]. Moreover, a decrease over 20 mmHg in systolic or over 10 mmHg in diastolic BP are also used as indicative of orthostatic incompetence [25].
- HR variability (HRV) measures used for diagnosis
HRV is determined as the variations in consecutive RR intervals and may be analyzed using long (usually 24 hour) or short (<10 min) ECG measurements, although the latter may be preferred for investigation purposes. Direct analysis provides HRV parameters in time domain, mainly standard deviations (SDNN) or root-mean squared deviations (RMSSD) of successive RR intervals [25]. Frequency-domain analysis is performed after data transform (e.g., fast Fourier transform) for obtaining HRV spectra. These histograms are divided in low (LF) and high frequency (HF) regions, which provide information on the influences of both autonomic branches on HR. The ratio LF/HF is mainly used as an index of autonomic balance, generally referred as sympathetic-to-parasympathetic ratio. Although a reduction in HF band may reflect a vagal dysfunction in early CAN, it may be accompanied by reduction in LF indicating concomitant dysregulation of sympathetic tone [25,26].
- Greco, C.; Santi, D.; Brigante, G.; Pacchioni, C.; Simoni, M. Effect of the Glucagon-Like Peptide-1 Receptor Agonists on Autonomic Function in Subjects with Diabetes: A Systematic Review and Meta-Analysis. Diabetes Metab. J. 2022, doi:10.4093/dmj.2021.0314.
- Vinik, A.; Erbas, T.; Pfeifer, M.; Feldman, E.; Stevens, M.; Russell, J. Diabetic autonomic neuropathy, 2004. In: The Diabetes Mellitus Manual: A Primary Care Companion to Ellenberg and Rifkin’s 6th Edition. Inzucchi SE, Ed. New York, McGraw Hill, 2004.
2) Are there any differences in the prevalence and characteristics of cardiac autonomic neuropathy between IDDM and NIDDM?
Nearly 10-20% patients with pre-diabetes and up to 10% recently diagnosed has associated confirmed CAN. The prevalence increases with age and disease duration, reaching 65% in long-term DM type 2 [1–3]. However, early diagnosis is difficult given the lack of characteristic signs of subclinical CAN and clinical investigation occurs only after the development of debilitating symptoms [4]. Age, disease duration and glycemic control are risk factors for development of CAN, while some controversy remains on sex, genetics and ethnicity. Metabolic syndrome and obesity are predictors of CAN, as indicated by correlations with body mass index, waist circumference, glucose tolerance, dyslipidemia, inflammation and oxidative stress [1,3]. Hypertension (treated or not), smoking and endothelial dysfunction accentuate disease progression and lead to worse prognosis [1–3,5].
- Spallone, V. Update on the Impact, Diagnosis and Management of Cardiovascular Autonomic Neuropathy in Diabetes: What Is Defined, What Is New, and What Is Unmet. Diabetes Metab. J. 2019, 43, 3–30, doi:10.4093/dmj.2018.0259.
2. Spallone, V.; Ziegler, D.; Freeman, R.; Bernardi, L.; Frontoni, S.; Pop-Busui, R.; Stevens, M.; Kempler, P.; Hilsted, J.; Tesfaye, S.; et al. Cardiovascular autonomic neuropathy in diabetes: clinical impact, assessment, diagnosis, and management. Diabetes. Metab. Res. Rev. 2011, 27, 639–653, doi:10.1002/dmrr.1239.
3. Ang, L.; Dillon, B.; Mizokami-Stout, K.; Pop-Busui, R. Cardiovascular autonomic neuropathy: A silent killer with long reach. Auton. Neurosci. 2020, 225, 102646, doi:10.1016/j.autneu.2020.102646. - Pop-Busui, R.; Boulton, A.J.M.; Feldman, E.L.; Bril, V.; Freeman, R.; Malik, R.A.; Sosenko, J.M.; Ziegler, D. Diabetic Neuropathy: A Position Statement by the American Diabetes Association. Diabetes Care 2017, 40, 136–154, doi:10.2337/dc16-2042.
- Bhati, P.; Alam, R.; Moiz, J.A.; Hussain, M.E. Subclinical inflammation and endothelial dysfunction are linked to cardiac autonomic neuropathy in type 2 diabetes. J. Diabetes Metab. Disord. 2019, 18, 419–428, doi:10.1007/s40200-019-00435-w.
3) A thorough review by a native English speaker would be helpful.
A native speaker colleague edited the manuscript.

Round 2
Reviewer 1 Report
Authors have addresses my concerns.